# Application of the Quality by Design Concept (QbD) in the Development of Hydrogel-Based Drug Delivery Systems

**DOI:** 10.3390/polym15224407

**Published:** 2023-11-14

**Authors:** S. Farid Mohseni-Motlagh, Roshanak Dolatabadi, Majid Baniassadi, Mostafa Baghani

**Affiliations:** 1School of Mechanical Engineering, College of Engineering, University of Tehran, Tehran 1439814151, Iran; 2Food and Drug Administration, Iran Ministry of Health and Medical Education, Tehran 1419943471, Iran

**Keywords:** drug delivery, hydrogel, quality by design, quality target product profile, critical quality attributes, critical material attributes, critical process parameter, design of experiment

## Abstract

Hydrogel-based drug delivery systems are of interest to researchers for many reasons, such as biocompatibility, high diversity, and the possibility of administration from different routes. Despite these advantages, there are challenges, such as controlling the drug release rate and their mechanical properties during the manufacturing of these systems. For this reason, there is a need for the production and development of such drug delivery systems with a scientific strategy. For this reason, the quality by design (QbD) approach is used for the development of drug delivery systems. This approach, by identifying the most effective factors in the manufacturing of pharmaceutical products and controlling them, results in a product with the desired quality with the least number of errors. In this review article, an attempt is made to discuss the application and method of applying this approach in the development of hydrogel-based drug delivery systems. So that for the development and production of these systems, according to the type of drug delivery system, what target characteristics should be considered (QTPP) and what factors, such as material properties (CMA) or process parameters (CPP), should be taken into account to reach the critical quality attributes of the product (CQA).

## 1. Introduction

Hydrogels are three-dimensional polymer network structures that are able to absorb large amounts of water [1]. They can be prepared through natural polymers, like alginate, chitosan, etc., or synthetic polymers such as polyvinyl alcohol (PVA), polyethylene glycol (PEG), etc., or a combination of natural and synthetic polymers simultaneously [2]. Hydrogels are usually insoluble due to chemical or physical crosslinks or chain entanglements [3]. They have emerged as one of the most promising candidates for biomedical applications due to their favorable mechanical properties, acute environmental sensing, biocompatibility, physical similarity to body tissue [4], and biodegradability characteristics [5]. One of these biomedical applications is drug delivery through hydrogel carriers. In recent years, hydrogels have gained considerable attention as drug carriers due to their ability to encapsulate and release a wide variety of therapeutic agents, including small molecules, proteins, and nucleic acids [5]. These capabilities make it possible to use hydrogels in dry or swollen states as drug carriers through oral, ocular, anal, injection, or other routes [6].

With all the advantages that hydrogel carriers have, there are also challenges in their manufacturing and drug loading, including controlled gelation temperature, controlled drug release rate from the hydrogel-based system [7], drug content uniformity in several similar systems, the good appearance of the drug product, the elasticity and mechanical strength of the hydrogel-based carrier [8], the time required for gelation, and many other such challenges. Therefore, to produce a hydrogel-based pharmaceutical system with proper performance, a reliable strategy to control all these challenges is required.

Today, the Quality by Design approach is used to manufacture drugs with the desired quality. Quality by Design (QbD) is a contemporary technique that streamlines product design and facilitates troubleshooting [9]. QbD is a strategy for designing and developing pharmaceutical formulations and manufacturing processes to align with predetermined product quality [10]. This concept examines all the parameters related to the production process and formulation characteristics in order to know the capacity of various changes that can affect product quality [9].

In order to study and develop a hydrogel-based pharmaceutical product using the QbD concept, one must know the elements and concepts of this approach. An ideal hydrogel-based drug delivery system should have a series of features known as quality target product profiles (QTPPs). QTPPs form the basis of design for product development [11]. For example, these target profiles include dosage form, drug administration route, drug release kinetics, stability, viscosity, drug solubility, and many other things [12]. From these QTPPs, a series of critical quality attributes (CQAs) are extracted that lead to product development. In fact, QTPPs are important criteria from the perspective of the patient and clinical outcomes, while CQAs are essential criteria from the point of view of the final quality and safety of the product, which leads to achieving the desired QTPPs [13]. These CQAs can include drug release characteristics, particle size, entrapment efficiency, pH, etc. In order to achieve the appropriate limits of these CQAs, the factors affecting them should be controlled, which can include characteristics related to materials known as critical material attributes (CMA) or related to the product manufacturing process known as critical process parameters (CPP). The CMAs can include, for example, the concentration and type of the surfactant, the concentration and type of the stabilizer, the concentration and type of crosslinker, or the ratio of the use of drugs and different polymers in the hydrogel-based drug delivery system. CPPs can also include things like temperature, sonication time, gelation time, etc. In fact, QbD explores the relationship between CMA/CPP and CQA, and by knowing these relationships, the product can be produced with the desired quality. Product development based on the concept of QbD includes a hierarchy of different elements (Figure 1).

Briefly, to make and develop a pharmaceutical product using the QbD concept, the target characteristics of the final product must first be listed (QTPPs), and then, according to these target characteristics, the critical quality attributes must be determined (CQAs). Using previous studies as well as guesses obtained based on experience, critical material attributes (CMAs) as well as critical process parameters (CPPs) that can affect CQAs should be listed. Because these effective factors may have a large number, some of them should be screened (Risk assessment). Using screening methods as well as risk assessment, a few of these factors are selected. Then, in order to identify their effect on CQAs, an experimental design should be created (DoE) by leveling these variables, and after the tests, using statistical analysis, the relationships between these variables and CQAs will be determined. Also, the design space, which shows the appropriate limits for each of the variables so that changes within these ranges do not cause significant changes in the final product, can be provided, and based on this, optimization can also be conducted. After this stage, based on the results of the research, control strategies for the continuous production of the pharmaceutical product should be determined.

In this review study, each element of the QbD approach has been addressed separately. In such a way that, for each element, first a general explanation of its nature and implementation method has been provided. Then, how the collected articles (studies related to hydrogel-based drug delivery systems) deal with each of these elements, briefly and as usefully as possible, is presented in the text and in the tables. An important point in conducting review studies is how to categorize the reviewed articles. The main category in this article is the route of drug administration and the drug delivery system. In this way, by knowing the route of drug administration and the closest drug delivery system to the system the reader wants to develop, he/she knows what input factors and responses should be considered. Also, the reader will know why each critical attribute of the final product was chosen. In this study, in addition to discussing the main elements of the QbD approach, more detailed issues such as scale-up studies and model-dependent analyses have also been discussed. According to our knowledge, there is no review article that has discussed the application of the QbD approach in studies that aim to study or develop a hydrogel-based drug delivery system. Also, in articles with topics close to this work, reviews from different angles, as seen in this work, have never been examined.

## 2. Quality by Design Elements

In the following, we will review each of the elements of the QbD approach in the collected articles.

### 2.1. Quality Target Product Profile (QTPP)

ICH Q8 defines QTPP as “a prospective summary of the quality characteristics of a drug product that ideally will be achieved to ensure the desired quality, taking into account the safety and efficacy of the drug product” [11]. At this stage, the targets for the desired medicinal product in terms of drug use in the clinical environment, route of administration, dosage form, dosage strength, drug stability, and other items are listed [14]. General and sometimes obvious goals are usually listed at this stage, and more detailed goals are discussed in the CQAs section. For example, the route of drug administration is one of the common QTPPs that is mentioned in almost all studies, while the drug maker definitely knows what route he plans to use for administration. Of course, more detailed matters are also discussed, such as appropriate limits of viscosity and pH or drug release, but often the targets that are determined are also general at this stage. For example, it is mentioned that the viscosity should be acceptable for the intended application or that the product should have a colored and smooth appearance [15]. While, for example, the appropriate range for viscosity or pH is usually discussed in the CQA section [16,17]. According to Table 1, cases like dosage form, route of administration, drug stability, dosage strength, or drug release characteristics are expressed in almost all hydrogel-based drug development studies using the QbD concept. But cases like odor, occlusivity, elasticity, or gelation time are rarely included in the list of QTPPs. Even cases like gelation temperature, which is an important feature in the studies of hydrogel-based drugs, are discussed in a few sources at this stage, and often these cases are discussed in the next sections. In the table below, a set of QTPPs that have been expressed in the studies of hydrogel-based drugs using the concept of QbD are presented. The interesting thing about the information in the table is that many different drug delivery systems and many kinds of drugs with different routes of administration have been developed using the QbD concept. This shows the high capacity of this concept in the development of hydrogel-based drug delivery systems.

### 2.2. Critical Quality Attributes (CQA)

Once the QTPPs are identified, the next step is to identify the corresponding CQAs [38]. Certainly, the determination of CQAs is one of the most important steps of a study based on the QbD approach. A CQA is defined as “a physical, chemical, biological, or microbiological property or characteristic that should be within an appropriate limit, range, or distribution to ensure the desired product quality” [11]. CQAs are subsets of QTPPs that may be affected by CMAs or CPPs. QTPPs, for example, include items such as dosage form or dosage strength, which do not change during the manufacturing process [35]. However, they also include items such as entrapment efficiency, drug release, particle size, viscosity, and many other items that can be affected by the formulation or manufacturing process and can be considered CQAs. For example, in the study conducted by Torregrosa et al. in order to develop an emulgel for the treatment of rosacea with the aid of the QbD approach, cases like drug strength, drug stability (at least 24), dosage form (emulgel), delivery system (bioadhesive oil in water emulgel), etc. exist among QTPPs, but they are not considered CQAs. while things like phase separation under mechanical stress or adhesion, which are on the list of QTPPs, are also considered CQAs [16]. These items can be changed by CMAs and CPPs. As mentioned in the definition of CQA, CQAs are characteristics that must be in a suitable range or distribution to achieve an optimal formula. This means that at this stage, apart from the detection of critical quality attributes, it is also important to determine the appropriate ranges for all or some of them. For example, in the study carried out by Garg et al. to achieve an optimal formulation of nanosized ethosome-based hydrogel formulations of Methoxsalen for enhanced topical delivery against vitiligo, the mass of phospholipid and ethanol were considered CMAs, and five responses such as vesicle size, percentage of drug entrapment (PDE), percentage of drug leakage (PDL), permeation flux (J), and skin deposition (SD) were considered CQAs. To achieve an efficient formula, the authors suggested an amount of 2–2.5 for PDL and 5.5–6.5 for J in the CQAs section. Also, the optimal value of PDE, vesicle size, and SD were considered as their maximum values. Finally, by conducting experiments and optimization, they were able to achieve a formula that has 2.16% phospholipid and 28.2% ethanol. The values of vesicle size, PDE, PDL, J, and SD were 281.3 nm, 67%, 2.12%, 5.8 µg/h/cm^2^, and 30.09 µg/cm^2^, respectively, and all of the values were consistent with their desired results [17].

In Table 2, for each active pharmaceutical ingredient (API) and type of drug delivery system, CQAs and their optimal ranges, the route of administration, and the reason for considering these CQAs are also mentioned. This information helps the reader produce these pharmaceutical systems or a system similar to them in an optimal state and also shows what responses of the pharmaceutical product should be tested and their justification. Also, the optimal value for each answer is roughly known. The reason that the route of administration is also mentioned in Table 2 and also in tables related to CPP and CMA, which are related to CMAs and CPPs, respectively, is that the route of administration, in addition to being a main categorizer of different drugs, can be very important in determining the critical quality attributes as well as the effective parameters in the production of pharmaceutical products. In general, it can be seen in Table 2 that although many and diverse systems have been investigated, it seems that for almost all hydrogel-based drug delivery systems, responses like particle size and entrapment efficiency are of special importance due to the effect of these responses on the amount of drug absorption.

When the CQAs are determined, it is time to find the factors that affect them. Then, by controlling them, we will reach our desired pharmaceutical product. In general, there are two categories of parameters affecting a CQA. CMAs and CPPs In the following, we will describe each of these cases.

### 2.3. Critical Material Attributes (CMA)

CMAs include the physicochemical, biopharmaceutical, or microbiological properties of the input materials, which guarantee the final quality of the product by being within the appropriate range. CMAs are the first group of factors that can cause variability in CQAs and are related to the composition of the drug product [12]. Some of the common CMAs in the production of hydrogel-based drug delivery systems include the concentration and type of surfactant, cosurfactant, drug, or polymer. In most of the studies conducted in this field, CMAs are considered more effective than CPPs in improving the quality of the final product. It is important to review this subject because the reader, knowing the drug delivery system, will know which CMAs have been found to be effective on the final quality of the drug in the literature and which CQA is most affected by this CMA.

Usually, the work process is that, first, a list of CMAs is prepared, and then, by performing the risk assessment stage, only a few of them are screened, which, according to the researcher’s opinion and based on previous studies, have the most effect on CQAs. For example, in the study conducted by Kang et al., to make and optimize tripterine-loaded nanostructures, the type of solid lipid, the type of liquid lipid, the ratio of solid to liquid lipid, the total mass of lipids, the type of surfactant, surfactant concentration, and drug mass were listed as primary CMAs, but only three factors, such as total lipid mass, surfactant mass, and drug mass, were considered independent variables that affect particle size, entrapment efficiency, and drug-loaded amount in the final product [44]. In some studies in the field of hydrogel-based drug delivery systems with the aid of the QbD approach, in addition to the drug and compounds, the hydrogel’s formula is of great importance. So, in some of these studies, a preliminary study in the case of hydrogel manufacturing based on QbD is also carried out. For example, in the study conducted by Durgun et al., for the purpose of optimizing the micellar-based in situ gelling system Posaconazole, 86 different compounds using poloxamer 407, poloxamer 188, and different grades of HPMC, MC, NaCMC, and Carbopol 980 (the first two polymers were considered the main gelling agents and the rest as auxiliary) were tested, and finally, based on the gelation temperature and other characteristics, the suitable compound was selected as the appropriate gel for the next stages of this study [27].

In Table 3, some CMAs that have been studied in the field of hydrogel-based pharmaceutical systems are mentioned as examples. The information in the table shows that according to the API and delivery system and the route of administration, what CMAs should be controlled in order to obtain a drug with the desired quality? It can be seen that the concentration and type of polymer in carriers and the amount of stabilizer or surfactant are among the most frequent CMAs in past studies.

As it is clear from the information in Table 3, the majority of studies of hydrogel-based pharmaceutical systems using QbD principles are about topically administered drugs. Usually, in this type of drug, CQAs include particle size, entrapment efficiency, PDI, drug release, or skin retention. Briefly, some of the CMAs affecting these CQAs in studies where the route of drug administration is topical include lipid concentration (in systems where lipids are also used as carriers) [22,41], surfactant concentration [18,22,41], and stabilizer concentration [23,45].

**Table 3 polymers-15-04407-t003:** Critical material attributes (CMA) for hydrogel-based drug delivery systems.

API and Delivery System	Route of Administration	CQA	CMA	Ref.
Capecitabine delivery by interpenetrating polymeric network (IPN) microbeads	Oral	-Particle size -Drug entrapment -Drug release	-Amount of polymer -Amount of cross-linker	[20]
Posaconazole loaded micellar based in situ gelling systems	Ocular	-Sol-gel temperature -Gelling capacity -Drug content -Log consistency index	-Poloxamer 188 (*w*/*v*%) -Poloxamer 47 (*w*/*v*%)	[27]
Apremilast-loaded solid lipid nanocarriers embedded in hydrogel	Topical	-Particle size -EE -PDI	-Lipid content (mg) -Surfactant concentration (%)	[22]
Hydrogel containing ketoconazole loaded cubosomes	Topical	-Particle diameter (nm) -PDI -Entrapment efficiency (%)	-Lipid (g) -Surfactant (mg) -Amount of stabilizer in 30 mL (% *w*/*w* of GMO)	[23]
Hydrogel formulation of econazole nitrate-loaded b-cyclodextrin nanosponges	Topical	-Particle size -Entrapment efficiency	-Polymer-linker ratio	[46]
Cinnarizine tablet using polyacrylamide-g-corn fiber gum	Oral	-Buoyancy lag time -Total buoyant duration	-Concentration of p-CFG (%) -Concentration of NaHCO_3_ (%) -Type of acid (%)	[47]
Dexamethasone sodium phosphate (DSP) and Tobramycin sulphate (TS) loaded thermoresponsive ophthalmic In Situ gel containing poloxamer 407 and Hydroxyl propyl methyl cellulose (HPMC) K4M	Ocular	-Gelation Temperatures -Gelation strength -Mucoadhesive Index -DSP Release in 9 h. -TS Release in 9 h.	-Concentration of Poloxamer 407(%) -Concentration of HPMC K4M (%)	[48]
Mupirocin-β-cyclodextrin complex loaded thermosensitive in-situ gel	Topical	-Gelation temperature (°C) -Adhesiveness (mJ)	-Pluronic F-127 (%) -Pluronic F-68 (%)	[49]
Antioxidant naringenin-loaded hydrogel for encouraging re-epithelization in chronic diabetic wounds	Topical	-Tensile strength (Mpa) -Swelling index (%)	-Na alginate concentration (%) -PVA concentration (%) -Pluronic F-127 concentration (%)	[50]

### 2.4. Critical Process Parameters (CPP)

CPPs are inputs related to the production process that have a direct and significant impact on critical quality attributes [51]. According to Table 4, some of the common CPPs in the production of hydrogel-based drug delivery systems include temperature [52], stirring speed [53], sonication time [22], and homogenization time [46]. For example, in the study conducted by Srivastava et al., which aimed to produce hydrogel formulations of econazole nitrate loaded in nanosponges, in addition to CMA, which is the polymer-linker ratio, stirring speed, homogenization speed, and homogenization time were considered as CPPs, and the effect of these factors on particle size and entrapment efficiency was investigated. Of course, these three CPPs were selected after risk assessment studies among factors such as temperature, heating time, homogenization speed and time, stirring speed, and beaker and bead sizes [46].

Reviewing the CPPs discussed in the literature is useful for the reader because, by knowing the drug delivery system, a good view of the effective factors related to the production process on the quality of the final product is provided, and by controlling them, an efficient drug delivery system can be obtained.

### 2.5. Risk Assessment and Factor Screening

In the risk assessment section, all possible CMAs and CPPs should be listed first (Table 5). Usually, for this purpose, researchers use the fishbone or Ishikawa diagram [59,60]. At this stage, studies may pay attention to the concept of 7 m (men, machines, material (API and excipients), measurement, methods, and milieu) in the selection of these variables affecting the CQAs of the product [59].

In a fishbone diagram, the desired quality product is usually considered the fish head. Among the effective factors (CPP/CMA), the more general items are considered large fishbones, and the subsets are considered smaller fishbones. For example, in Figure 2, which is related to the study of Rapalli et al., conducted to investigate the behavior of solid lipid nanocarriers embedded in hydrogel for topical delivery of apremilast, it can be seen that the effect of the material, which is a CMA itself, is written on one of the large bones of the fish, and subsets like gelling agent, HLB, and type of surfactant play the role of smaller fish bones. The effect of these factors on characteristics (CQA) such as entrapment efficiency, PDI, or zeta potential is investigated [22]. After that, it is necessary to select the most effective parameters by applying a method. This is because, firstly, by reducing the number of effective parameters, it will be easier to interpret the results in the continuation of this work, and the number of experiments can increase exponentially with the addition of each variable. The risk assessment matrix (RAM) method [59] or the risk estimation matrix (REM) [18] are usually used in the studies.

The process of this work is that in a matrix similar to Table 6, the impact of each parameter on each CQA is determined by three levels of impact (“Low” impact, “Medium” impact, and “High” impact) [59]. Then, all or some of these factors are graded based on three indicators of severity (S), occurrence (O), and detectability (D), and usually, the multiplication of these three factors is considered an RPN (Risk Priority Number) (Table 7). Scoring these factors is based on prior knowledge and a literature review [31]. Finally, factors with higher RPN are selected for further work (the design of the experiment) [30]. In some other studies, in order to screen as many factors as possible with a higher impact, a regression relationship is first created by considering CQAs as dependent variables and entering all CMAs and CPPs as independent variables. Finally, all the variables that have a significant effect are screened for the continuation of this study [19,46,61]. Of course, in some studies, some variables may be examined in the pre-study stage (preliminary) and not enter the risk assessment stage [16,62]. In some other studies, the screening stage of the factors may not be provided at all, and they go directly to the stage of the design of the experiment. For example, in the study conducted by Desu et al. to investigate a superporous hydrogel composites-based gastroretentive drug delivery system to investigate the effect of different factors on the two responses of swelling rate and drug release, without listing several input factors (CMAs and CPPs) and screening them, PVA mass, Glutaraldehyde percentage, and Span 80 percentage were selected as variables affecting the final quality of the product to conduct QbD-based studies [45]. In Table 5, a number of studies are given as examples of what parameters were screened after the risk assessment process. The information in the table helps to know other effective factors (CPP/CMA) on each drug delivery system response (CQA) in addition to the factors used in the experiment design stage. It can be seen that usually about three factors are selected among many factors.

**Table 5 polymers-15-04407-t005:** All the input and screened factors for the hydrogel-based drug delivery systems.

API and Delivery System	Input Factors (CMA/CPP)	Screened Factors	Ref.
Resveratrol loaded ethosomal hydrogel	1 Type of phospholipid	1 Concentration of Phospholipid 2 Concentration of ethanol	[21]
	2 Concentration of phospholipid (%)		
	3 Type of aqueous phase		
	4 Volume of aqueous phase (mL)		
	5 Concentration of ethanol (%)		
	6 Stirring speed (rpm)		
	7 Stirring Time (min)		
	8 Type of stirrer		
	9 Method of preparation		
	10 Sonication speed and time		
Phospholipid microemulsion-based hydrogel for enhanced topical delivery of lidocaine and prilocaine	1 Conc. of IPM (Isopropyl myristate)	1 conc. of oil (IPM) 2 conc. of Smix (A combination of nonionic surfactant (Tween 80) and saturated lipids (SL) (Labrasol/Lauroglycol 90) and Phospholipon 90 G: ethanol) 3 conc. of water	[32]
	2 Conc. of Tween 80		
	3 Conc. Of Labrasol/Lauroglycol 90		
	4 Ratio of PL: Ethanol		
	5 Water		
	6 ME (microemulsions) stirring time		
	7 ME stirring speed		
	8 Temperature		
Lidocaine and prilocaine loaded-nanoemulsion system	1 Emulsifier conc. (% *w/w*)	1 Emulsifier concentration (g) 2 Homogenisation pressure (bar) 3 Homogenisation cycle (cycle)	[56]
	2 Homogenisation pressure (bar)		
	3 Poloxamer conc. (% *w/w*)		
	4 Phospholipid conc. (% *w/w*)		
	5 Primary emulsion stirring speed (rpm)		
	6 Primary emulsion stirring time (min)		
	7 PG conc. (% *w/w*)		
	8 Oil conc. (% *w/w*)		
	9 Homogenisation cycles		
	10 Type of oil		
	11 Type of emulsifier		
Apremilast-loaded solid lipid nanocarriers embedded in hydrogel	1 Amount of Lipid	1 Lipid (mg) 2 % Surfactant 3 Sonication time (minutes)	[22]
	2 Surfactant concentration		
	3 Sonication Time		
	4 Stirring speed		
	5 Stirring Time		
	6 Temperature of surfactant solution		
Resveratrol-loaded mucoadhesive lecithin/chitosan nanoparticles for prolonged ocular drug delivery	1 Lecithin concentration	1 Lecithin concentration 2 Drug concentration	[63]
	2 chitosan to lecithin ratio		
	3 PF127 concentration		
	4 organic solvent		
	5 organic to aqueous ratio		
	6 injection rate		
	7 stirring speed		
	8 needle type		
	9 lecithin grade		
	10 MW of chitosan		
	11 Deacetylation degree of chitosan		

### 2.6. Design of Experiment (DoE)

After screening the variables, it is necessary to create an experimental design (DoE). To develop a product of commercially acceptable quality, it is important to choose an appropriate experimental design. A proper experimental design can provide benefits such as time savings and cost-effective production [64]. Usually, methods such as Taguchi [65], factorial [66], mixture, or response surface are used at this stage. After conducting the experiments according to the created design, statistical analyses are performed on the results obtained from the experiments, and then the design space is obtained according to the desired ranges for each response. In fact, the design space provides values or levels of independent variables that can best meet the qualitative needs of the product for which we have created the design of the experiment [67]. The advantage of the design space is that when working within the design space ranges, changes are not considered significant changes [15]. This means that for each independent variable, we will have ranges, and if the input values are within these ranges, the final product will be obtained with the manufacturer’s desired quality. The introduction of these limits causes more freedom of choice in the drug production phase. Because, regarding the values of the input factors (CPP/CMA), instead of fixing them at a specific value, it is possible to choose the value within the range. Few studies present these ranges explicitly. But in many studies, this work is presented using overlay plots [68,69,70]. In this way, according to Figure 3, one of the factors is placed on the vertical axis and one on the horizontal axis, and the curves related to the answers are drawn in the figure along with their values. These curves for each answer show the values of the first and second factors that will result in a constant value for that answer. Different answers (single or multiple responses) are drawn on these graphs. Usually, by examining the intersection points of response curves (in graphs where multiple responses are shown on one graph), the design space area is shown in a different color (usually yellow) (Figure 3) [61,71]. After this stage, optimization is usually provided within the design space, which can provide the best values of the independent variables (to achieve the desired answers). For example, in Singh et al.’s study conducted in order to achieve optimal composition for gastroretentive bilayer tablet lamivudine and zidovudine, for each layer of the tablets (both lamivudine layer and zidovudine layer), different combinations (CMAs) of CP971 and HPMC polymers (each at three levels) were tested to obtain desired results for five answers (CQAs), such as time taken for 60% drug release (T60%), amount of drug release in 16 h (Q16h), diffusion drug release exponent (*n*), buoyancy time (Tb), and bioadhesive strength (BS). After performing 13 runs for each layer and obtaining relationships between independent and dependent variables, response surfaces and design space were presented graphically. In such a way that for each layer, by knowing the mass of the polymers, it was possible to have the value of each of the answers by two and three-dimensional diagrams. Also, the region where, by choosing the mass of polymers within those ranges, the optimal limits of responses would be obtained was also graphically specified. Also, with the maximization of Tb, BS, n, Q16h, and T60% for the prognosis of the optimized formulation, the optimal combinations were also presented, and the obtained values were within the specified limits [62]. In many studies, there is not much talk about the design space. But an important result that is presented in most studies and is almost one of the most important results of a study based on the concept of QbD is the optimal value of input variables. In Table 8, a number of studies that have performed optimization are presented. It can be seen that every study using an experimental design tries to achieve the optimal values of the input factors to achieve the desired answers.

#### Relation between Independent and Dependent Variables

One of the results that is often presented in the study of hydrogel-based drug delivery systems using the QbD approach is the relationships between independent and dependent variables, so that the reader can obtain the value of the dependent variables by knowing the values or levels of the independent variables. This is different from what happens in the design space and optimization sections, because in some studies, the design space or optimization is presented, but these relationships are not available in the article [15]. This usually happens when the researcher uses artificial neural networks (ANNs) to optimize and discover relationships. But when methods such as response surfaces are used, relationships between input and output variables are presented. In these models, all or some of the CPPs or CMAs or their functions are considered independent variables, and all or some of the CQAs or their functions are considered dependent variables. Then, the results are presented using statistical modeling. For example, in the research conducted by Kang et al., three answers, including particle size, entrapment efficiency, and drug load, based on three variables such as total lipid content (mg), surfactant content (mg), and amount of added TRI (mg), were predicted, and their equations were presented using the quadratic model [44]. Usually, multiple linear regression [19], quadratic [21], or 2FI [40] models are used to build these models, and of course, these models are usually associated with goodness of fit above 0.9. It shows the good correlation between input factors and responses. In general, the presentation of such models can be of great help to the readers in reproducing pharmaceutical products according to the formulations in the articles. In Table 9, several articles that have explicitly presented the relationships between factors and answers are presented. It can be seen that in each article, what relationship was used to create relationships between what responses (CQA) and what independent variables (CMA/CPP), as well as the values of goodness of fit, are presented.

### 2.7. Control Strategy in Order to Continuous Production

After the optimization and design space, it is time for the “control strategy” stage to control the continuous drug production process. ICH guideline Q10 on pharmaceutical quality systems defines the control strategy as “a planned set of controls, derived from current product and process understanding, that assures process performance and product quality. The controls can include parameters and attributes related to drug substance and drug product materials and components, facility and equipment operating conditions, in-process controls, finished product specifications, and the associated methods and frequency of monitoring and control” [80].

This stage provides the necessary basic knowledge for the product manufacturing stage. In fact, the results of the previous section specify the control strategy for the production process and ensuring the quality of the pharmaceutical product [19]. That is, this stage has the task of ensuring that the product continuously maintains the desired quality during the production process, and it is based on the results of the previous section [12,81]. The upper and lower limits for CQAs, CPPs, and CMAs can be defined as the control space (or normal operating range) where these parameters are typically controlled during production to ensure repeatability [61]. There is an important point when determining the limits of the control space based on the design space. The limits of the control space, although they are located within the limits of the design space, should not be much smaller than them. Because in this case, the control of the process becomes very complicated [61].

During the “control strategy” stage, the input materials, the characteristics of the pharmaceutical product, and the production units should be monitored so that the final product always maintains its quality in the production process [12,81]. In this section, a lot of attention is usually paid to the concept of Process Analytical Technology (PAT). In fact, PAT provides the means to achieve QbD goals. The purpose of the PAT concept in the pharmaceutical industry is to provide quantitative and qualitative information to monitor and control the production process, as well as optimize and make efficient use of energy, time, and raw materials. That is, the main task of the PAT concept is process monitoring. Tools such as spectroscopy techniques, NIR sensors, Raman spectroscopy, and terahertz pulse spectroscopy are used to implement PAT [82]. The reason that there is much less information in this field is that most articles are silent about this step. This could be due to the fact that many studies do not reach the stage of mass production at all, and those that do often do not publish information on how to implement control strategies.

#### Scale-Up Study

In some studies, in addition to the usual elements of the QbD approach, after the optimization stage, scale-up studies are also conducted [27,36,39,41,55]. Scale-up is an important concept in the pharmaceutical industry and biopharmaceutical production. Scale-up studies try to take the biopharmaceutical production process from a laboratory scale to a commercial scale [83].

The reason why we have addressed this section separately is that even among the drugs that show very promising results in the stages of laboratory studies, very few of them have the ability to be produced as a commercial product in the market [64]. Also, one of the most important research and development processes in the pharmaceutical industry is scale-up studies to produce large and numerous batches for clinical and pre-clinical studies. Therefore, the presence of scale-up studies in an article definitely increases the credibility of this work. In a few articles, scale-up studies have been conducted. For example, in the study by Mahmood et al. conducted in order to optimize Luliconazole-loaded lyotropic liquid crystalline nanoparticles for topical delivery, after optimizing and identifying the optimal values for the formulation (A: Lipid content; B: Surfactant content; C: co-surfactant content), a scale-up study was conducted by augmenting the selected LUL dispersion (F-3) to 50 mL for that batch, and the results were relatively close to what was obtained in the previous step. The optimization was conducted by minimizing the particle size and maximizing the entrapment efficiency [41]. In another study conducted by the same authors to achieve an optimal composition and also to characterize the LUL drug loaded in NLCs, by changing the variables (X1: amount of lipid—X2: concentration of surfactant—X3: sonication time) and testing 17 batches, finally minimizing the globule size and maximizing the entrapment efficiency, they found the optimal batch and conducted scale-up studies by augmenting that batch to 100 mL [55].

## 3. Model Dependent

In many studies that have studied the development of a hydrogel-based pharmaceutical system using the QbD approach, the kinetics of drug release have also been discussed (Table 10). As it has been seen before, in some studies, the release characteristics of the drug itself have been considered a CQA [84]. Therefore, it is definitely important to check that the drug release from each drug delivery system is similar to that of the famous drug release models. Usually, to check this, well-known models such as Zero-order, First-order, Higuchi, and Korsmeyer–Peppas are tested, and the most suitable model is presented [18,20,40,48].

## 4. Conclusions

For the development of a drug delivery system, it is very common to use the quality by design (QbD) concept. By using this concept, it is possible to produce pharmaceutical products with minimal errors and optimal quality by controlling effective input parameters. In recent years, this approach has been widely used for the development of hydrogel-based drug delivery systems. In this study, an attempt was made to investigate the use and application of the QbD approach in the development of these systems. It was observed that this approach is used for the development of various types of delivery systems, including in situ gels, injectable gels, semi-solid systems, NLCs, etc., as well as various administration routes such as oral, topical, ocular, etc. Among the critical quality attributes (CQA) that are important for these types of systems, drug release rate, entrapment efficiency, and particle size are the most frequent features. The characteristics of materials (CMA), such as the concentration of various polymers such as HPMC, Poloxamer, or other polymers (based on the application), in the production of hydrogel carriers are very effective in improving the quality of the final product. In addition to the characteristics related to materials, process parameters (CPP) such as sonication and homogenization time (based on the application) should also be controlled to obtain a product of ideal quality. Although many studies have investigated hydrogel-based drug delivery systems and their critical input and output factors have been introduced, the scarcity of studies that have brought these results to scale is quite evident. Also, the vast majority of articles ignore the final stages of the QbD approach, including the control strategy, while it is not possible for the continuous production of pharmaceutical products with the quality achieved in the laboratory studies stage without adopting a suitable control strategy. Therefore, discussing the details of the control strategy and how to implement it to obtain a quality product during continuous production will be an interesting and new topic for researchers.

## Figures and Tables

**Figure 1 polymers-15-04407-f001:**
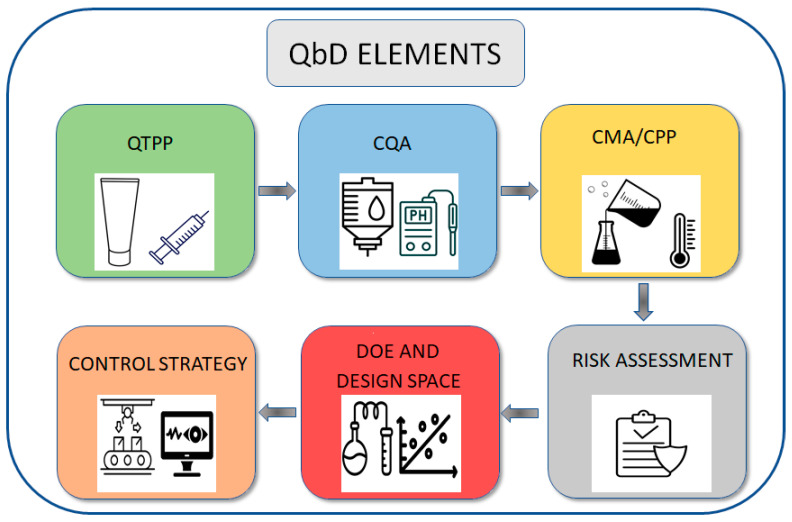
Quality by design (QbD) elements are available online at vecteezy.com (accessed on 23 September 2023).

**Figure 2 polymers-15-04407-f002:**
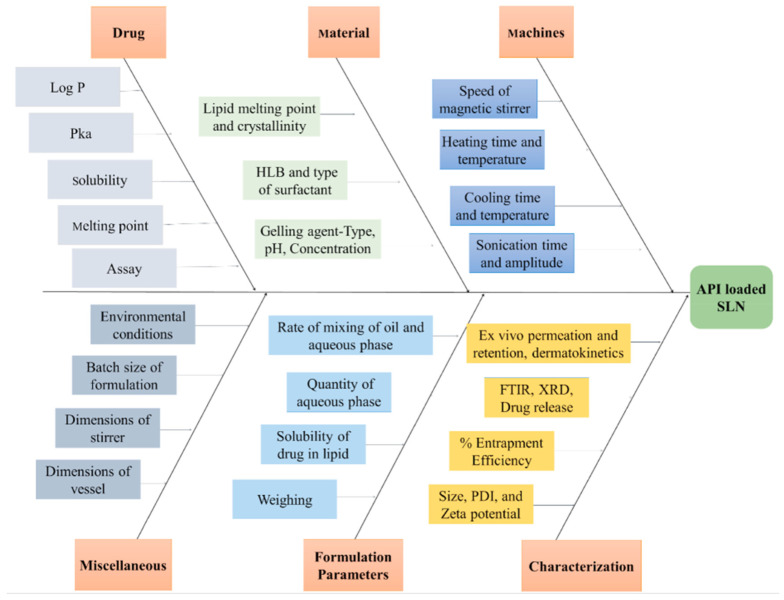
An example of the Fishbone diagram [22].

**Figure 3 polymers-15-04407-f003:**
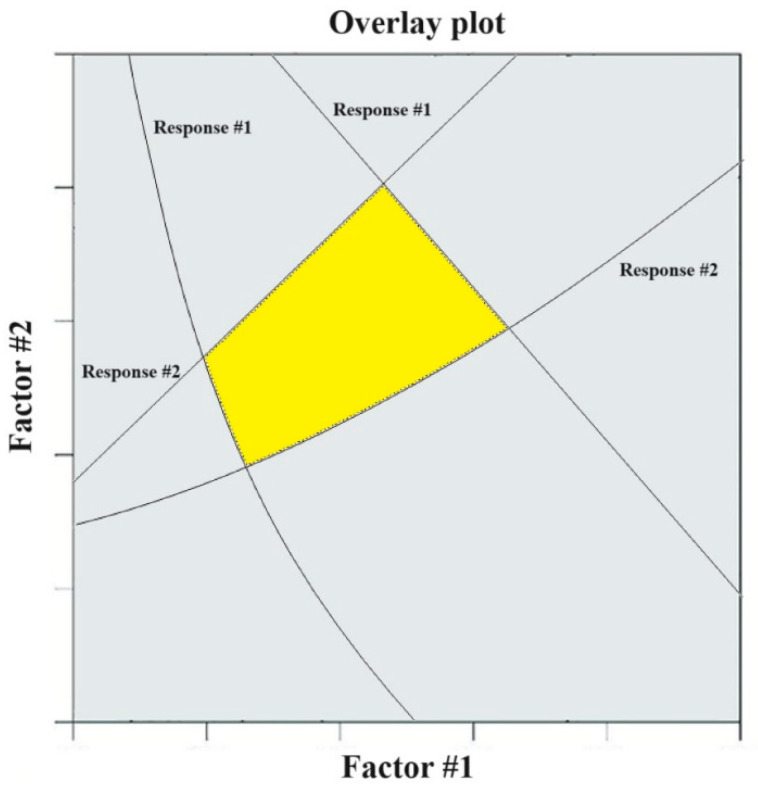
The optimized area is marked in yellow.

**Table 1 polymers-15-04407-t001:** Different quality target product profiles (QTPPs) for hydrogel-based drug delivery systems.

QTPP	Target
Dosage form/Delivery system	SMEDDS incorporated into gel [18]—Semisolid system [17]—Emulgel [16]—Hydrogel [19]—IPN hydrogel microbeads [20]—Ethosomes [21]—Solid lipid nanocarriers dispersion loaded gel [22]—Cubosome dispersion loaded gel [23]—Semisolid [24]—Microemulsion [25]—NS [26]—Micellar-based in situ gels [27]—Cream [28]—Injectable hydrogel [29]-
Route of administration	Topical [16,19,21,22]—oral [20]—Dermal [24]—Nasal [25]—Ocular [27]—Intratumoral [29]-
Dosage strength	0.75% metronidazole; 4% niacinamide [16]—5 g/100 g (5%) lidocaine [17]—0.05% *w*/*w* apremilast [22]—0.2% *w*/*w* ketoconazole [23]—15 mg aripiprazole [25]-
Stability	At least 24 [16]—At least 6 months at various storage temperatures [21,25]
Container closure system/Packaging	Collapsible opaque tube [16,30]—Container closure system qualified as suitable for this drug product [31]—Aluminium tubes [32]—Amber glass c
Method of administration	Once or twice daily application [16]
Appearance/Clarity	colored smooth emulgel [16]—White smooth textured gel [22]—White smooth textured gel [23]—Clear [27]—Transparent Gel [30]—Transparent [33]-
Odor	No unpleasant odor [16,31]
Dosage type/Dosage design/Drug release	Controlled drug release [15]—Rapid release [21,25]—Sustained release [22]—Control release compared to free drug [23]—Modified release [24]—More than 80% [26]
Viscosity	Acceptable spreadability for application on the lesion [15]
Occlusivity	Higher occlusivity [15]—Maximum [24]
Method of preparation	Ionotropic gelation technique [20]
Skin permeation	Higher flux and skin retention [21,30]
Particle size	<200 nm [22]—<300 nm [23]
Entrapment/Encapsulation efficiency	Maximum [22]—The maximum entrapment helps in improved permeation [23]—>%95 [24]—High [34]
Healing effect/Therapeutic indication/Therapeutic use	Anti-cellulite [24]—Skin allergies [35]- local anesthetic [17]—Colon cancer [20]—Anti-inflammatory [36]
pH	Between 6 and 7 [24]—6.6–7.8 [27]—Compatible with skin [34]
Log consistency index	0–1 [27]
Tsol-gel temperature/Gelation temperature	30–35 °C [27]—37–45 °C [33]
Gelling capacity/Gelationg time	Gelation immediately [27]— 10–30 min [37]
PDI	Uniform [34]—≤0.3 [28]
Elasticity	Elastic and flexible [33]—Medium [37]

**Table 2 polymers-15-04407-t002:** Critical quality attributes (CQA) for hydrogel-based drug delivery system.

API and Delivery System	Route of Administration	CQAs and Their Targets	Justification	Ref.
Apremilast-loaded lyotropic liquid crystalline nanoparticles embedded hydrogel	Topical	-Particle size (Minimum< 200 nm) -Entrapment (Maximum)	-Entrapment: The higher the entrapment, the higher the skin permeation and skin retention -Particle size: The smaller the particle size, the greater the occlusive effect and improved permeation	[39]
Gelatin-based hydrogel patch	Topical	-Lubricant properties 0.005–0.05 N·m -Elasticity (flexible 0.3–1.7 mm) -Adhesiveness (0.3–0.8) N -Gelation temperature 37–45 °C	-Gelation temp: Maintain integrity when applied to the skin -Lubricant properties: To hold the mask on the face and decrease pressure -Elasticity: To avoid patch breakage Adhesiveness To fix the patch on the skin	[33]
THermosensitive hydrogel loaded with IgY and LL37-SLNs	Topical	-Sol-gel temperature (In range (30–35 °C)) -Sol-gel time (Minimum) -Drug release (Continuous release in 72 h)	-Sol-gel temp: Controlled within the oral temperature range to facilitate in situ coagulation -Sol-gel time: Reduced waiting times, thereby improving patient adherence -Release kinetic: Sustained drug release is expected to prolong the anti-bacterial action of the drug at targeted gingival sulcus	[40]
Itraconazole-loaded micro-emulsionbased hydrogel	Topical	-pH (5–7) -Viscosity (500–700 × 103 cps) -Globule size (Minimize the upper (≤100 nm) limits)	-Globule size: Globule size is a key control parameter for product permeation and retention of the drugs within the dermal layers. The smaller the globule size, the better the permeation of the drugs. Moreover, globule size has a direct effect on product appearance for the patient and physician -Viscosity: Drug retention in the skin is another important property of MBH formulations, which governs the ability of the formulation to behave like a reservoir and provides adequate drug residence in the skin; hence, viscosity should be critically considered -Ph: Skin irritation at the delivery site should be identified during the development of the delivery systems. Skin irritation reactions are usually caused by direct exposure to the skin. The features of topical and transdermal systems, such as formulation components (APIs, permeation enhancers, and other excipients), occlusion of the skin, and duration of the delivery systems, could cause skin irritation. Thus, pH could be considered a response factor in optimizing formulation	[31]
Quercetin—SMEDDS incorporated into gel	Topical	-% Transmittance (>90%) -Globule size (100–150 nm) In vitro Release (>90% at end of 8 h) -% Permeate (Higher skin retention)	-% Transmittance: % Transmittance is a critical and fundamental attribute in the formulation of SMEDDS, as it represents the optical birefringence and homogeneity of the formulation, thereby affecting the efficacy of SMEDDS -Particle size: Particle size affects the physicochemical and drug release properties and is considered a benchmark for the stability of the formulation. Smaller globule sizes allow better solubility, a higher surface area, and better permeation at the therapeutic site; hence, they are regarded as highly critical -In vitro release: In vitro release is an important parameter that will determine the availability of the drug at the site of action, thereby affecting safety and efficacy, hence being considered a critical CQA -% Permeate: The product under development is for local action at the site of the wound. In order for higher efficacy, increased retention is required at the wound site	[18]
Resveratrol-loaded polymeric micelles-based carbomer gel	Topical	-Globule size (In range (100–200 nm)) -Micellar incorporation efficiency (MIE) (Highest) -The extent of Resveratrol deposition in the dermal layer of skin (Skin deposition, SD) (High)	-Globule size: It was considered highly critical due to its importance in the permeation and retention of the bioactive in the dermal layer. Smaller sizes facilitate movement inside the layers of skin, but beyond a certain level, it leads to systemic absorption -MIE: Higher incorporation efficiency is required to reduce the quantity of formulation to be applied and the pharmaceutical properties of the formulation -Skin deposition: High skin retention is required for better therapeutic benefits because the target site is located in the dermis region of the skin. Therefore, it was also selected as highly critical	[30]
Luliconazole-loaded lyotropic liquid crystalline nanoparticles	Topical	-Particle size (nm) (minimize (<200 nm)) -Entrapment Efficiency % (maximum)	-Particle size: The small size of the LCNP ensures close contact of lipidic particles with the lipid bilayer of the stratum corneum, leading to increased penetration of drug particles into the deeper layers of the skin -Entrapment Efficiency: High entrapment efficiency will translate into better therapeutic efficacy	[41]
Delivery of lidocaine and prilocaine by phospholipid microemulsion-based hydrogel	Topical	-Globule size (Less than 100 nm) -Flux (high) -Cumulative drug permeation(high) -Skin retention (high)	-Globule size: A smaller globule size of ME will facilitate better permeation and retention of the drugs within the dermal layers; hence, it was regarded as highly critical -Flux: Essential parameter to access the topical delivery potential of the formulation for enhanced therapeutic efficacy. Thus, it was considered highly critical -Cumulative drug permeation: The permeation property of the drug formulations is highly responsible for attaining meaningful pharmacodynamic effects; hence, it was taken up as highly critical -Skin retention: High skin retention is important for topical anesthesia, wherein the pain-sensitive nerve endings (target site) are located in the dermis region of the skin. Hence, it was regarded as highly critical.	[32]
Capecitabine delivery by interpenetrating polymeric network (IPN) microbeads	Oral	-Particle size (Size range around 500 μm) -Drug entrapment (Minimum 60–90%) -Drug release (Extended)	-Particle size: Particle size is a critical parameter, as the large microbeads were non-spherical, worm-shaped, and had an uneven surface -Drug entrapment: Indicator of high drug entrapment -Drug release: IPN delivery vehicles are meant to release the drug for an extended period of time	[20]
Polymeric nanospheres of terbinafine hydrochloride for topical treatment of onychomycosis using a nano-gel formulation	Topical	-Particle size (<250 nm) -PDI (<0.3) -Recovery (Maximum possible) -Zeta potential (>40 mV)	-Particle size: Suitable for effective permeability -PDI: Impacts physical stability and drug uniformity -Recovery: Ensures formulation efficiency and supports the desired drug release -Zeta potential: It helps with dispersion stability and particle uniformity	[42]
RSV nanoemulsions are dispersed into a hydrogel matrix, assembling nanoemulgels (NEGs).	Topical	-Droplet diameter (<200 nm) -7 day size change of droplet (<10%) -pH (4.5–6)	-Droplet diameter: Droplet size < 200 nm is optimal for macrophage uptake -7 days size change of droplet: Nanodroplets must remain stable throughout their time of use -pH To maximize hydrogel viscosity, pH needs to make at least wounds less susceptible to chronic infection when the wound environment is slightly acidic, normal skin pH 5–6	[43]

**Table 4 polymers-15-04407-t004:** Critical process parameters (CPP) for hydrogel-based drug delivery system.

API and Delivery System	Route of Administration	CQA	CPP	Ref.
Solid lipid nanocarriers embedded in hydrogel for topical delivery of apremilast	Topical	-Particle Size -Entrapment efficiency -PDI	-Sonication time	[22]
Hydrogel formulation of econazole nitrate-loaded b-cyclodextrin nanosponges	Topical	-Particle size -Entrapment efficiency	-Stirring speed -Homogenization time -Homogenization speed	[46]
CS loaded optimized AgN-CA gel (Microwave-assisted)	Topical	-Particle size -Absorbance	-Power of microwave in Watt	[54]
Luliconazole-loaded nanostructured lipid carriers (NLCs)	Topical	-Particle size -Entrapment efficiency	-Sonication time	[55]
Lidocaine and prilocaine-loaded nanoemulsion system	Topical	-Particle size -PDI	-Homogenization pressure -Homogenization cycle	[56]
*n*-Propyl gallate encapsulated solid lipid nanoparticle-loaded hydrogel for intranasal delivery	Intranasal	-Average hydrodynamic diameter (Z-average) -Polydispersity index (PDI) -Zeta potential	-Temperature at dissolution phase	[52]
Chitosan-ca-alginate microspheres for colon delivery of celecoxib hydroxypropyl-b-cyclodextrin-PVP complex	Oral	-Entrapment efficiency -Drug released after 4 h in colonic medium	-Time of cross-linking	[57]
Transdermal delivery of phytosomal Manjistha extract gel (MJE gel)	Topical	1 Vesicular size 2 Entrapment efficiency 3 PDI	-Rotation speed	[58]

**Table 6 polymers-15-04407-t006:** Initial Risk Assessment Matrix.

CQA	CMA #1	CMA #2	…	CMA #n	CPP #1	CPP #2	…	CPP #*n*
CQA #1	Low/Medium/High	Low/Medium/High	Low/Medium/High	Low/Medium/High	Low/Medium/High	Low/Medium/High	Low/Medium/High	Low/Medium/High
CQA #2	Low	Medium	High	Low/Medium/High	Low/Medium/High	Low/Medium/High	Low/Medium/High	Low/Medium/High
…	Low/Medium/High	Low/Medium/High	Low/Medium/High	Low/Medium/High	Low/Medium/High	Low/Medium/High	Low/Medium/High	Low/Medium/High
CQA #*n*	Low/Medium/High	Low/Medium/High	Low/Medium/High	Low/Medium/High	Low/Medium/High	Low/Medium/High	Low/Medium/High	Low/Medium/High

Sometimes factors with low, medium, and high risk impacts are colored green, yellow, and red, respectively.

**Table 7 polymers-15-04407-t007:** Risk priority number assessment.

**CMA**	**S (Severity)**	**O (Occurrence)**	**D (Detectability)**	**RPN**	**Impact on CQA**
CMA #1	1-n	1,2,…,n	1,2,…,n	1,2,…,n^3^	CQA #*n*, CQA #m
CMA #2	1,2,…,n	1,2,…,n	1,2,…,n	1,2,…,n^3^	CQA #*n*, CQA #m
…	1,2,…,n	1,2,…,n	1,2,…,n	1,2,…,n^3^	CQA #*n*, CQA #m
CMA #*n*	1,2,…,n	1,2,…,n	1,2,…,n	1,2,…,n^3^	CQA #*n*, CQA #m
CPP #1	1,2,…,n	1,2,…,n	1,2,…,n	1,2,…,n^3^	CQA #*n*, CQA #m
CPP #2	1,2,…,n	1,2,…,n	1,2,…,n	1,2,…,n^3^	CQA #*n*, CQA #m
…	1,2,…,n	1,2,…,n	1,2,…,n	1,2,…,n^3^	CQA #*n*, CQA #m
CPP #*n*	1,2,…,n	1,2,…,n	1,2,…,n	1,2,…,n^3^	CQA #*n*, CQA #m

Usually, n is equal to 10 or 100. Usually, a limit is considered for RPN, and values higher than that are considered effective factors. In some studies, the impact on the CQA column, like this table, is presented.

**Table 8 polymers-15-04407-t008:** Optimized values of input factors (CMA/CPP) and responses (CQA) with an applied experimental design.

API and Delivery System	Optimized Values of CMAs and CPPs (Unit) [Value]	Obtained Optimal Values for CQAs (Unit) [Value]	Applied Experimental Design	Number of Runs	Ref.
Luliconazole-loaded Microemulgel	1 Oil (%) [6.03] 2 Smix (%) [55.9]	1 Transmittance (%) [99.23] 2 Viscosity (cps) [145.12] 3 CDR (%) (cumulative drug release after 9 h.) [25.91]	Full factorial	9	[72]
Alginate-Chitosan Nanoparticles of Simvastatin	1 Chitosan (g) [0.258] 2 Sodium Alginate (g) [0.353]	1 Particle Size (nm) [142.56] 2 Entrapment efficiency (%) [75.18]	Central Composite Design (RSM)	13	[73]
Neomycin Sulfate Gel Loaded with Solid Lipid Nanoparticles	1 Stearic acid (%) [0.467] 2 Glycerol monostearate (%) [0.275] 3 P-F 68 (%) [1.23]	1 Particle size (nm) [196.25] 2 EE (%) [89.27]	Box–Behnken design (RSM)	17	[74]
Diclofenac-loaded ethosomes	1 Ethanol concentration (%) [22.9] 2 Phosphatidylcholine: Cholesterol ratio (%) [88.4:11.6]	1 Vesicle size (nm) [144 ± 5] 2 Zeta potential (mV) [23.0 ± 3.76] 3 Elasticity [2.48 ± 0.75] 4 Entrapment efficiency (%) [71 ± 4]	Full factorial	20	[75]
Chitosan-Ca-alginate microspheres for colon delivery of celecoxib hydroxypropyl-b-cyclodextrin-PVP complex	1 Alginate (%) [3.9 for systemic therapy] [4.5 for local therapy] 2 CaCl2 (%) [7.2 for systemic therapy] [11 for local therapy] 3 Chitosan (%) [0.5 for systemic therapy] [2.6 for local therapy] 4 Time of cross-linking (min) [12 for systemic therapy] [18.5 for local therapy]	1 Entrapment efficiency (%) [90.0 ± 2.9 for systemic therapy] [65.0 ± 2.9 for local therapy] 2 Drug released after 4 h in colonic medium (%) [99.6 ± 2.1 for systemic therapy] [25.0 ± 2.5 for local therapy]	Doehlert	23	[57]
Emulsion-based nano tailored gel for improved antiphotoageing potential of Silymarin	1 Oil (%) [10.0] 2 Smix (%) [36.67] 3 Water (%) [51.33]	1 Globule size (nm) [100.0] 2 Cumulative drug release (%) [83.59]	D-optimal mixture design	14	[76]
Pioglitazone (PZ) encapsulated in a carbopol-based transgel system (proniosomes/niosome)	1 Tween 80 (mg) [150.0] 2 Phospholipid (mg) [45.0] 3 Cholesterol (mg) [25.0]	1 Vesicle size (nm) [426.85] 2 Entrapment efficiency (%) [91.46] 3 Transdermal flux (µg/cm^2^/h) [50.84]	Box–Behnken design (RSM)	15	[77]

**Table 9 polymers-15-04407-t009:** Type and goodness of fit of relationships between independent (CMA/CPP) and dependent (CQA) variables.

CQA	CMA/CPP	Relation Type	Goodness of fit	Ref.
1 Globule size (nm)	1 amount of TTO (g)	1 Quadratic	0.9957 (R^2^)	[78]
2 In vitro release (%)	B, amount of tween 80 (g)	2 Quadratic	0.9976 (R^2^)	
1 EE	1 S-protected TO (STO)	1 Quadratic	0.9738 (Adjusted R^2^)	[53]
2 In vitro mucoadhesion	2 Stirring Speed	2 Quadratic	0.9797 (Adjusted R^2^)	
1 Sol-gel temperature (°C)	1 Poloxamer 407 (P407) 2 Poloxamer 188 (P188)	1 Quadratic	1 0.9942 (R^2^)	[40]
2 Solgel time (s)		2 2FI	2 0.9100 (R^2^)	
3 Drug release (%)		3 Quadratic	3 0.9995 (R^2^)	
1 Vesicle size	1 Concentration of Phospholipid 2 Concentration of ethanol	1 Quadratic	1 0.9656 (R^2^)	[21]
2 Entrapment efficiency		2 Quadratic	2 0.9902 (R^2^)	
3 Permeation flux J		3 2FI	3 0.9612 (R^2^)	
4 Skin deposition SD		4 Quadratic	4 0.9648 (R^2^)	
1 MDDC particle size (D10)	1 Rotation speed 2 Stirrer type used for organic solvent removal	1 2FI	1 0.9312 (R^2^)	[19]
2 MDDC particle size (D10)		2 Quadratic	2 0.9566 (R^2^)	
3 MDDC particle size (D10)		3 Linear	3 0.7693 (R2)	
4 MDDC particle size distribution (span)		4 2FI	4 0.8852 (R^2^)	
1 Vesicle size	1 Soya lecithin (Phospholipon 90 G) concentration	1 Quadratic	1 0.98 (R^2^)	[79]
2 PDI	2 Ethanol concentration	2 2FI	2 0.95 (R^2^)	
3 Entrapment efficiency	3 Stirring speed	3 Quadratic	3 0.97 (R^2^)	
1 Particle Size	1 Tween 80 concentration	1 Quadratic	1 0.999 (R^2^)	[77]
2 Entrapment	2 Phospholipid concentration	2 Quadratic	2 0.998 (R^2^)	
3 Flux	3 Cholesterol concentration	3 Quadratic	3 0.997 (R^2^)	

**Table 10 polymers-15-04407-t010:** Model-dependent analyses of drug release from hydrogel-based drug delivery systems.

API and Delivery System	Fitted Model	Ref.
Luliconazole loaded Microemulgel	Higuchi	[85]
In situ gelling microemulsion of Lorazepam	Korsmeyer–Peppas	[86]
Apremilast-loaded lyotropic liquid crystalline nanoparticles embedded hydrogel	first-order	[39]
Luliconazole-loaded nanostructured lipid carriers (NLCs)	First-order	[55]
Luliconazole-loaded lyotropic liquid crystalline nanoparticles	Higuchi	[41]
Diclofenac-loaded lyotropic liquid crystal nanoparticles	First-order	[36]
Octreotide loaded peptide-based hydrogel	Weibull	[87]

## Data Availability

All data obtained during this study are available from the corresponding author on reasonable request.

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
