# Peer review of "Application of the Quality by Design Concept (QbD) in the Development of Hydrogel-Based Drug Delivery Systems"

_polymers, 2023, doi:10.3390/polym15224407_

Round 1

Reviewer 1 Report

Comments and Suggestions for Authors

The creation and optimization of targeted drug delivery systems, as the basis of personalized medicine, is one of the trending topics in modern materials science.

The authors devoted their review to the wood field, namely to optimize the design of hydrogel-based targeted delivery systems. It is always difficult to evaluate and re-evaluate review articles on such a broad topic, so the amount of information and articles is enormous and constantly updated.

First of all, it is worth noting the quantity and quality of the quoted material. The total number of sources is 94 articles and reviews, of which only 24 (25%) relate to publications older than 5 years. That is, from the point of view of the relevance of the information discussed in the work, there is no doubt.

Usually in their papers, the authors publish good characteristics such as critical quality attributes and target quality profile of the product. Because the focus is on targeted outcomes, such as toxicity reduction and targeted targeting. However, from an applied point of view, these parameters are not enough for further development of the project.

The authors have spent a large amount of time and effort to systematize these parameters, which will move the application aspect forward.

In the "Critical Material Attributes" section, they devoted a lot of attention to the major applications of data delivery and how it is introduced into the body. However, there is no information about such popular and basic materials in the field of creating hydrogels as agarose and polyacrylamides.

However, the reviewer, as the author, understands that it is impossible to cover the entire list of possible topics and materials, so this deficiency is not considered critical.

Otherwise, there are no quality complaints about the review. And on the whole it deserves to be accepted for publication as it is.

Author Response

Greetings and Regards
 We understand that you have carefully read the material in the article and we are honored by your kind comments. Thank you very much for your good opinion about our work. We are also very happy with your high understanding of the challenges in writing. 
With renewed respect

Reviewer 2 Report

Comments and Suggestions for Authors

This manuscript is an interesting and nice summary of the quality by design concept in hydrogel-based drug delivery systems. It discusses each element of the quality by design and lists current studies as examples. It would be nice for the authors to discuss how to evaluate each element and how these analysis could help design the next-generation hydrogel-based drug delivery system.

Comments on the Quality of English Language

The manuscript needs to be carefully proofread. 

Reviewer 3 Report

Comments and Suggestions for Authors

In this review, authors investigated the use and application of the QbD approach in the development of delivery systems, including in situ gels, injectable gels, semi-solid systems, NLCs, etc., as well as various administration routes such as oral, topical, ocular, etc. Among the critical quality attributes (CQA) that are important for these types of systems, drug release rate, entrapment efficiency, and particle size are of the most frequent features. The topic is interesting, but some points should be revised before the final decision of the paper.

1.       Authors described different elements of the QbD approach, however, the relationship among them was not discussed.

2.       When using acronyms, authors need to follow relevant norms to ensure the readability and understandability of the paper. In general, when a word appears less than three times in the text, it is not necessary to define an abbreviation, just spell out the full name. The number of occurrences should be calculated according to the abstract, text, note and table respectively. The acronyms like CQA, CAM, CPP, etc., are confusing.

3.       The keywords should also be revised. There are too many keywords in the current version. The language in some places are hard to read. Moderate editing of English language required.

4.       The review has no figures with scientific meanings. There are two figures named figure 1.

5.       Authors also discussed the influence of QbD approach on various administration routes such as oral, topical, ocular, etc. Why? The logic herein is not obvious.

6.       Several recent papers on hydrogels can be cited and compared: https:// doi: 10.1016/j.ijbiomac.2018.04.123.

Comments on the Quality of English Language

Moderate editing of English language required
